# Binary Independent Component Analysis: A Non-stationarity-based Approach

**Antti Hyttinen**[1,2]  **Vitória Barin-Pacela**[1,2,3]  **Aapo Hyvärinen**[1]

[1]Department of Computer Science, University of Helsinki, Helsinki, Finland
[2]Helsinki Institute for Information Technology, Finland
[3]Mila, Université de Montréal, Montréal, Canada

## Abstract

We consider independent component analysis of binary data. While fundamental in practice, this case has been much less developed than ICA for continuous data. We start by assuming a linear mixing model in a continuous-valued latent space, followed by a binary observation model. Importantly, we assume that the sources are non-stationary; this is necessary since any non-Gaussianity would essentially be destroyed by the binarization. Interestingly, the model allows for closed-form likelihood by employing the cumulative distribution function of the multivariate Gaussian distribution. In stark contrast to the continuous-valued case, we prove non-identifiability of the model with few observed variables; our empirical results imply identifiability when the number of observed variables is higher. We present a practical method for binary ICA that uses only pairwise marginals, which are faster to compute than the full multivariate likelihood. Experiments give insight into the requirements for the number of observed variables, segments, and latent sources that allow the model to be estimated.

## 1 INTRODUCTION

Despite significant progress in both linear and nonlinear ICA in recent years [Hyvärinen and Morioka, 2016, Hyvärinen et al., 2019, Khemakhem et al., 2020], ICA for binary data remains a challenging and important problem as binary data is abundant in various fields, such as bioinformatics, health informatics, social sciences, natural language, and electrical engineering. An ICA model for binary data may also open new opportunities in solving problems closely related to ICA, such as causal discovery [Shimizu et al., 2006] and feature extraction [Hyvärinen and Morioka, 2016].

Methods for binary ICA have been proposed based on either binary or continuous-valued independent components. In the case of binary components, Himberg and Hyvärinen [2001] and Nguyen and Zheng [2011] assumed an OR mixture model. In addition, some extensions of Latent Dirichlet can be seen as binary ICA [Podosinnikova et al., 2015, Buntine and Jakulin, 2005]. On the other hand, Kabán and Bingham [2006] presented an approach based on a latent linear model and binarized observations, although the components were restricted to the unit interval, which limits its applicability. Recently, Khemakhem et al. [2020] presented a nonlinear ICA model (iVAE) that can employ binarized observations, making several contributions that we can build on.

Our goal is to study the prospects of ICA for binary data using a model that is both theoretically analyzable and intuitively appealing. It is crucial to investigate the identifiability of such a model, and to have a consistent estimator which is not based on approximations whose validity are not clear. None of the approaches above fulfills all of these criteria.[1]

We propose a binary ICA model inspired by recent developments in nonlinear ICA. We formulate a latent linear model with a separate binarizing measurement equation. Crucially, we assume the components to be non-stationary, which is a powerful principle and very useful here because any non-Gaussianity (commonly employed in ICA) may be destroyed by binarization. Thus, we obtain a binary ICA model whose likelihood can be described in closed-form via the multivariate Gaussian cumulative distribution function. We further propose to combine the likelihood with a moment-matching approach to obtain a fast and accurate estimation algorithm. In fact, due to the model structure, pairwise marginal distributions of non-binarized data can be accurately estimated from the binary data and the likelihood can be computed directly from them. We investigate the identifiability of the model, and somewhat surprisingly, we show that low-dimensional models are in fact non-identifiable—

---

[1]As noted in the Corrigendum of Khemakhem et al. [2020] (v4 on arXiv), their initial identifiability proof for a discrete non-linear ICA model is incorrect.

*Accepted for the 38th Conference on Uncertainty in Artificial Intelligence* (UAI 2022).

while higher-dimensional models are (empirically) shown to be identifiable.

# 2  A MODEL FOR BINARY ICA

In this section, we define a binary counterpart of the linear ICA model. In particular, we consider here a model based on non-stationarity of the components, and start by motivating such an approach.

## 2.1  THE APPROACH OF NON-STATIONARITY

While often non-stationarity is considered a nuisance, in the theory of ICA it is well-known that a suitable non-stationarity of the independent components can be very useful. Pham and Cardoso [2001] already used it in the case of linear ICA, and Hyvärinen and Morioka [2016] extended the idea to nonlinear ICA. Note that the mixing is assumed stationary, and the non-stationarity is a statistical property of the components only.

In line with such literature, we assume the $n$-dimensional data is divided into $n_u$ segments which express the non-stationarity, i.e. the segments have different distributions. In the case of time series, we may be able to find such segmentation simply by taking time bins of equal sizes. Such non-stationarity based on a segment-wise (piece-wise stationary) model is well-known in linear ICA [Pham and Cardoso, 2001, Miettinen et al., 2017]. Formally, each data point has a segment index $u$ assigned to it.

In fact, this setting is more general and it is not necessary to have time-series. The additionally "observed" variable $u$ makes the non-stationarity a special case of the auxiliary variable framework of Khemakhem et al. [2020]. It is thus not only natural in the case of non-stationary time series, but also when there is any other external discrete variable, such as the experimental condition or intervention, or even a class label that modulates the distribution of the data.

The motivation for such a non-stationary model is that it can greatly extend the identifiability of ICA. Linear ICA is identifiable if the components are simply non-Gaussian, which is why the utility of non-stationarity in that context has always been dubious and such algorithms are rarely used. However, in the case of *non*linear ICA, non-Gaussianity does not enable identifiability, which may be intuitively clear since a nonlinear transformation can change the marginal distributions quite arbitrarily from non-Gaussian to Gaussian or vice versa. A major advance was in fact obtained by Hyvärinen and Morioka [2016], who showed that non-stationarity does enable identifiability in the nonlinear case.

Here, we propose that using non-stationarity of the components is very useful in the case of binary data as well. Again, intuitively, non-Gaussianity is likely to be rather useless

since the binarization destroys any detail about the non-Gaussianity of the distributions, and such a model would be unlikely to be identifiable. However, non-stationarity is *not* destroyed by binarization. Thus, binary ICA can be estimated based on non-stationarity of the components, as we will show later in this paper.

## 2.2  FORMAL MODEL DEFINITION

To define the model in detail, we assume the $n$-dimensional data is generated from $n_z$ latent variables (independent components, or sources), collected into a latent random vector $\mathbf{z}^u$, which are generated independently of each other from a Gaussian distribution. Crucially, the parameters of the Gaussian distribution change as a function of the segment as

$$\mathbf{z}^u \sim \mathcal{N}(\boldsymbol{\mu}_{\mathbf{z}}^u, \boldsymbol{\Sigma}_{\mathbf{z}}^u)$$

where $\boldsymbol{\Sigma}_{\mathbf{z}}^u$ is a diagonal matrix of the source variances in segment $u$.

We define "intermediate" variables $\mathbf{y}^u$ which are a linear mixing of the sources by a mixing matrix $\mathbf{A}$ with $n$ rows and $n_z$ linearly independent columns

$$\mathbf{y}^u = \mathbf{A}\mathbf{z}^u \sim \mathcal{N}(\mathbf{A}\boldsymbol{\mu}_{\mathbf{z}}^u, \ \mathbf{A}\boldsymbol{\Sigma}_{\mathbf{z}}^u\mathbf{A}^\intercal). \tag{1}$$

Here the mixing matrix $\mathbf{A}$ is constant, i.e., stationary, over the segments $u$ [Pham and Cardoso, 2001].

While some work in ICA considers noisy continuous observations by adding noise to $\mathbf{y}^u$, we can consider here binarized observations $\mathbf{x}^u$ instead. The binarization is done using a linking function $\sigma$ so that the probability of $i$th element of $\mathbf{x}^u$ being 1 is:

$$P(x_i^u = 1) = \sigma(y_i^u).$$

We use a linking function based on the Gaussian CDF (cumulative distribution function):

$$\sigma(y_i^u) = \Phi\left(\sqrt{\frac{\pi}{8}} y_i^u \Big| 0, 1\right)$$

where $\Phi$ is the cumulative distribution function of the Gaussian distribution, here with mean 0 and variance 1. We use $\sqrt{\pi/8}$ as the coefficient to match closely to the sigmoid function $\sigma(y_i) = \frac{1}{1+e^{-y_i}}$ [Waissi and Rossin, 1996, Li, 2021], which is standardly used in statistics and machine learning in similar linking contexts.

We directly allow for different coefficients instead of $\sqrt{\pi/8}$, but our estimation methods assume that the linking function has the particular form. The motivation is to allow for closed-form expressions of the Gaussian integrals involved in Section 3 in terms of the Gaussian CDF. The difference to the logistic function is very small, while the methods are much simpler with the used linking function. In fact, our

ICA model allows for closed-form likelihood with this particular linking function (Section 3), which would be difficult to achieve with a logistic linking function.

Furthermore, the linking function has the following intuitive interpretation. Take $y_i^u$, add independent noise $\epsilon$ from $\mathcal{N}(0, \frac{8}{\pi})$, and binarize $y_i^u$ simply by a hard threshold 0 to get $x_i^u$. This gives the same distribution for $x_i^u$, since the probabilities match:

$$P(x_i^u = 1) = P(y_i^u + \epsilon > 0) = P(\epsilon > -y_i^u)$$
$$= \int_{-y_i^u}^{\infty} \mathcal{N}\left(\epsilon \middle| 0, \frac{8}{\pi}\right) d\epsilon = \Phi\left(\sqrt{\frac{\pi}{8}} y_i^u \middle| 0, 1\right).$$

A binary ICA model $\mathcal{M} = (\mathbf{A}, \{\boldsymbol{\mu}_{\mathbf{z}}^u\}_u, \{\boldsymbol{\Sigma}_{\mathbf{z}}^u\}_u)$ thus consists of the following parameters: the mixing matrix $\mathbf{A}$, the means $\boldsymbol{\mu}_{\mathbf{z}}^u$ and the diagonal (co)variance matrices $\boldsymbol{\Sigma}_{\mathbf{z}}^u$ for all segments $u$, denoted by $\{\boldsymbol{\mu}_{\mathbf{z}}^u\}_u$ and $\{\boldsymbol{\Sigma}_{\mathbf{z}}^u\}_u$. Consequently, it defines a distribution for a binary vector $\mathbf{x}^u$ in each segment indexed by $u$.

## 3 THE LIKELIHOOD

A surprising observation regarding the the latent variable model defined in Section 2 is that we can calculate the likelihood in closed-form by employing the multivariate Gaussian CDF. For example, the model defines the probability of the data vector of all ones, denoted by $\mathbf{1}$, as:

$$P(\mathbf{x}^u = \mathbf{1}|\mathcal{M}) = \int P(\mathbf{x}^u = \mathbf{1}|\mathbf{y}^u) P(\mathbf{y}^u|\mathcal{M}) d\mathbf{y}$$
$$= \int \Phi\left(\sqrt{\frac{\pi}{8}} \mathbf{y}^u \middle| \mathbf{0}, \mathbf{I}\right) \mathcal{N}(\mathbf{y}^u | \mathbf{A}\boldsymbol{\mu}_{\mathbf{z}}^u, \mathbf{A}\boldsymbol{\Sigma}_{\mathbf{z}}^u \mathbf{A}^{\mathsf{T}}) d\mathbf{y}$$

where the univariate Gaussian CDFs are written as a multivariate Gaussian CDF $\Phi$ with an identity covariance matrix. The benefit of using a Gaussian CDF-based linking function comes into play here, as the value of the integral is directly a value of a multivariate Gaussian CDF [Waissi and Rossin, 1996, Li, 2021]: The above formula actually specifies the probability of first drawing $\mathbf{y}^u$, multiplying it by $\sqrt{\pi/8}$, and then, independently, drawing a standard Gaussian variable $\mathbf{n} \sim \mathcal{N}(\mathbf{0}, \mathbf{I})$ that is element-wise smaller. We therefore have:

$$P(\mathbf{x}^u = \mathbf{1}|\mathcal{M}) = P\left(\mathbf{n} - \sqrt{\frac{\pi}{8}} \mathbf{y}^u < \mathbf{0}\right)$$

This motivates us to define a random vector $\mathbf{q}^u$, an important construct in the following developments, as:

$$\mathbf{q}^u = \mathbf{n} - \sqrt{\frac{\pi}{8}} \mathbf{y}^u, \qquad (2)$$

which is simply a noisy, re-scaled and sign-flipped version of the linear mixture $\mathbf{y}^u$. In fact, since $\mathbf{q}^u$ is the sum of

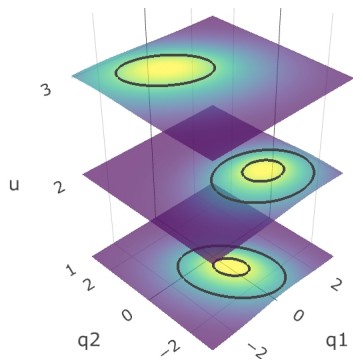

Figure 1: Binary ICA model for two observed variables and three segments. For each segment, there is a bivariate Gaussian distribution on $\mathbf{q}^u$, the probability of an assignment to the binary observed variables is the probability mass in the corresponding quadrant.

two independent Gaussian random vectors, it also has a Gaussian distribution $\mathbf{q}^u \sim \mathcal{N}\left(\boldsymbol{\mu}_{\mathbf{q}}^u, \boldsymbol{\Sigma}_{\mathbf{q}}^u\right)$ with:

$$\boldsymbol{\mu}_{\mathbf{q}}^u = -\sqrt{\frac{\pi}{8}} \mathbf{A}\boldsymbol{\mu}_{\mathbf{z}}^u, \qquad (3)$$
$$\boldsymbol{\Sigma}_{\mathbf{q}}^u = \mathbf{I} + \frac{\pi}{8} \mathbf{A}\boldsymbol{\Sigma}_{\mathbf{z}}^u \mathbf{A}^{\mathsf{T}}. \qquad (4)$$

The probability of the data vector of ones in segment $u$ is, then:

$$P(\mathbf{x}^u = \mathbf{1}|\mathcal{M}) = P\left(\mathbf{q}^u < \mathbf{0}\right) = \Phi\left(\mathbf{0}|\boldsymbol{\mu}_{\mathbf{q}}^u, \boldsymbol{\Sigma}_{\mathbf{q}}^u\right), \qquad (5)$$

where the cumulative distribution function of the multivariate Gaussian $\Phi$ has all variables integrated from $-\infty$ to 0; it is readily implemented in basic packages [Genz and Bretz, 2009].

Similar derivation gives the probabilities for other assignments to $\mathbf{x}^u$. These probabilities can be expressed compactly for all value assignments as:

$$P(\mathbf{x}^u|\mathcal{M}) = \Phi\left(l(\mathbf{x}^u), u(\mathbf{x}^u)|\boldsymbol{\mu}_{\mathbf{q}}^u, \boldsymbol{\Sigma}_{\mathbf{q}}^u\right) \qquad (6)$$

in which the multivariate Gaussian probability density function is integrated from the lower bound $l(\mathbf{x}^u)$ to the upper bound $u(\mathbf{x}^u)$, with the $i$th elements in the bounds defined by:

$$l(\mathbf{x}^u)[i] = \begin{cases} -\infty \text{ if } x_i^u = 1 \\ 0 \text{ otherwise} \end{cases} \quad u(\mathbf{x}^u)[i] = \begin{cases} 0 \text{ if } x_i^u = 1 \\ \infty \text{ otherwise} \end{cases}$$

Importantly, this formulation allows for a particularly clear intuitive interpretation of the model. Figure 1 shows this for two observed variables and three segments. For each segment, the model defines a bivariate Gaussian distribution for $\mathbf{q}^u$, depicted by colors and contours on the planes. The probability for an assignment of the observed binary variables $\mathbf{x}^u$ in a segment is simply the probability mass in a

corresponding quadrant. The multivariate Gaussian distributions for $\mathbf{q}^u$ in each segment are related in the sense that they are formed by the same mixing matrix performing on independent sources particular to the segment.

The log-likelihood of the whole data set can then be calculated as

$$l \;\; = \;\; \sum_u \sum_{\mathbf{x}^u} c(\mathbf{x}^u) \log \Phi(l(\mathbf{x}^u), u(\mathbf{x}^u) | \boldsymbol{\mu}_{\mathbf{q}}^u, \boldsymbol{\Sigma}_{\mathbf{q}}^u), \;\; (7)$$

where $c(\mathbf{x}^u)$ is the count of the data points with assignment $\mathbf{x}^u$ in a segment $u$ and the sum is taken over all assignments to $\mathbf{x}^u$ and $u$.

# 4 ON IDENTIFIABILITY

Many ICA models can only be identified up to scaling and permutation indeterminacies of the sources [Hyvärinen et al., 2001, Khemakhem et al., 2020]. Straightforwardly we can see that those limitations apply for our model as well. By re-ordering columns of the mixing matrix and the sources, the implied distribution is unaffected; similarly, we can counteract the scaling (or sign-flip) of the mixing matrix columns by scaling (or sign-flipping) the sources. However, binarization actually induces additional indeterminacies as we will show next.

## 4.1 THE BINARIZATION INDETERMINACY

Recall that the probability of an assignment to binary $\mathbf{x}^u$ is given by the probability of the Gaussian $\mathbf{q}^u$ landing in different regions (Equation 5). But note that the probability in Equation 5 stays exactly the same even if $\mathbf{q}^u$ is multiplied by a diagonal matrix $\mathbf{Q}^u$, possibly different for each segment $u$, with positive entries (scaling factors) on the diagonal:

$$P\left(\mathbf{q}^u < \mathbf{0}\right) \;\; = \;\; P\left(\mathbf{Q}^u \mathbf{q}^u < \mathbf{0}\right).$$

This is valid even if the elementwise operator is $>$ or a mixture of $>$ and $<$.[2] Figure 2 shows an example of this equivalence relation for one segment and two observed variables. The two Gaussian distributions for $\mathbf{q}^u$ represented by the blue and red contours imply the exact same joint distribution for binary observed variables $\mathbf{x}^u$. The amount of mass in each of the 4 quadrants is exactly the same. This means that we essentially lose all scale information on $\mathbf{q}^u$ in the binarization.

Then, two binary ICA models $\mathcal{M} = (\mathbf{A}, \{\boldsymbol{\mu}_{\mathbf{z}}^u\}_u, \{\boldsymbol{\Sigma}_{\mathbf{z}}^u\}_u)$ and $\hat{\mathcal{M}} = (\hat{\mathbf{A}}, \{\hat{\boldsymbol{\mu}}_{\mathbf{z}}^u\}_u, \{\hat{\boldsymbol{\Sigma}}_{\mathbf{z}}^u\}_u)$ are indistinguishable if

---

[2]For the probability of $\mathbf{x}^u$ being all ones, any permutation matrix $\mathbf{Q}^u$ would similarly preserve the implied probability, but the probability of some other assignment for $\mathbf{x}$ (each of which corresponds to some mixture of $>$ and $<$) may change then.

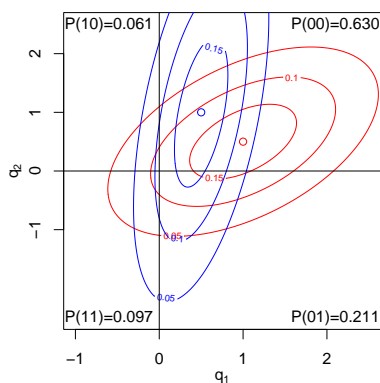

Figure 2: Two Gaussian distributions (red and blue) for a two dimensional $\mathbf{q}^u$ which imply the same binary distributions after binarization by the linking function. That is because the mass of both distributions in each of the 4 quadrants is identical.

there are positive diagonal matrices $\{\mathbf{Q}^u\}_u$ such that for each segment $u$, the means and covariances of $\mathbf{q}^u$ satisfy:

$$\hat{\boldsymbol{\mu}}_{\mathbf{q}}^u \;\; = \;\; \mathbf{Q}^u \boldsymbol{\mu}_{\mathbf{q}}^u, \qquad (8)$$
$$\hat{\boldsymbol{\Sigma}}_{\mathbf{q}}^u \;\; = \;\; \mathbf{Q}^u \boldsymbol{\Sigma}_{\mathbf{q}}^u \mathbf{Q}^u, \qquad (9)$$

which can be written more clearly using the model parameters (Equations 3 and 4) as:

$$\sqrt{\frac{\pi}{8}} \hat{\mathbf{A}} \hat{\boldsymbol{\mu}}_{\mathbf{z}}^u \;\; = \;\; \mathbf{Q}^u \sqrt{\frac{\pi}{8}} \mathbf{A} \boldsymbol{\mu}_{\mathbf{z}}^u, \qquad (10)$$
$$\mathbf{I} + \frac{\pi}{8} \hat{\mathbf{A}} \hat{\boldsymbol{\Sigma}}_{\mathbf{z}}^u (\hat{\mathbf{A}})^\mathsf{T} \;\; = \;\; \mathbf{Q}^u (\mathbf{I} + \frac{\pi}{8} \mathbf{A} \boldsymbol{\Sigma}_{\mathbf{z}}^u \mathbf{A}^\mathsf{T}) \mathbf{Q}^u. \;\; (11)$$

This limits identifiability possibilities (Section 4.2) but nevertheless also allows for the development of efficient estimation procedures in Sections 4.3 and 5.2.

## 4.2 THE ROW ORDER INDETERMINACY

One of the consequences of the binarization indeterminacy is the following non-identifiability result concerning $n = 2$ observed variables, proven in Appendix A in the supplement.

**Theorem 1.** *If the row order of the 2-by-2 mixing matrix $\mathbf{A}$ of a binary ICA model is reversed, then the source means $\boldsymbol{\mu}_{\mathbf{z}}^u$ and variances $\boldsymbol{\Sigma}_{\mathbf{z}}^u$ can be adjusted such that the implied distributions for the observed binary $\mathbf{x}^u$ remain identical.*

This means that in addition to column order and scale, we also have row order indeterminacy here. Although the result may generalize to certain sparse higher dimensional models, fortunately, it does not jeopardize the estimation of higher dimensional models in general.

This result does have consequences for causal discovery [Shimizu et al., 2006, Suzuki and Inaoka, 2021, Peters

et al., 2011, Inazumi et al., 2014]. Consider two structural equation models, implying opposite causal directions:

$$\mathbf{y}^u = \begin{pmatrix} 0 & 0 \\ b & 0 \end{pmatrix} \mathbf{y}^u + \mathbf{z}^u, \quad \mathbf{y}^u := \begin{pmatrix} 0 & b \\ 0 & 0 \end{pmatrix} \mathbf{y}^u + \mathbf{z}^u.$$

where $\mathbf{z}^u$ has a Gaussian distribution in each segment $u$ with diagonal covariance matrix $\mathbf{\Sigma}_{\mathbf{z}}^u$. The models correspond respectively to the mixing models (compare to Equation 1):

$$\mathbf{y}^u = \begin{pmatrix} 1 & 0 \\ b & 1 \end{pmatrix} \mathbf{z}^u, \quad \mathbf{y}^u = \begin{pmatrix} 1 & b \\ 0 & 1 \end{pmatrix} \mathbf{z}^u.$$

If we observed binarized $\mathbf{y}^u$, i.e. $\mathbf{x}^u$, we can at most identify the mixing matrix up to row order, column order and column scale. By switching the column order and then the row order of the mixing matrix on the left, we get the mixing matrix on the right. Thus, unlike in the continuous case, we cannot detect the causal direction between two variables without further limiting assumptions or information on other variables.

## 4.3 THE CORRELATION IDENTIFIABILITY

Note that the indistinguishable models satisfying Equation 9 or Equation 11 have equal *correlation matrices* (i.e. matrices of Pearson correlation coefficients) for the random variables $\mathbf{q}^u$. The next theorem and corollary show that the correlations between elements of $\mathbf{q}^u$ are indeed theoretically identifiable from the distributions of the binary observed variables $\mathbf{x}^u$. Intuitively, the higher the correlation, the more likely will the pair of binary observed variables in $\mathbf{x}^u$ receive equal assignments. The fairly technical proof is given in Appendix B in the supplement.

**Theorem 2.** *Two binary ICA models imply different distributions for binary observations $\mathbf{x}^u$ (in a given segment $u$) if the correlation matrices for $\mathbf{q}^u$ are not equal.*

This result is crucial for the development of our novel estimation method (Section 5.2), via the corollary:

**Corollary 1.** *The correlation matrix of $\mathbf{q}^u$ in a given segment $u$ is identifiable from the distribution for binary $\mathbf{x}^u$.*

On the other hand, the following theorem recaps the well-known result [Hyvärinen et al., 2001, Pham and Cardoso, 2001] that the means do not help in estimating the mixing matrix (proven in Appendix C):

**Theorem 3.** *If two models $\mathcal{M}$ and $\hat{\mathcal{M}}$ with $n = n_z$ imply the same correlation matrices for $\mathbf{q}^u$ (in a given segment) then the means $\boldsymbol{\mu}_{\mathbf{z}}^u$ can be adjusted such that the implied binary distributions are identical.*

## 5 METHODS FOR BINARY ICA

Next, we present three methods for estimating the binary ICA model, building on the theory in Sections 3 and 4. The BLICA method of Section 5.2 is the main novel algorithmic contribution of the paper.

## 5.1 MAXIMUM LIKELIHOOD ESTIMATION

We have already derived the likelihood of the binary ICA model in Equation 7. A straightforward approach is then to optimize this using e.g. L-BFGS [Liu and Nocedal, 1989]. The gradient involves the moments for the *truncated* multivariate Gaussian distribution, which can be obtained from R package tmvtnorm [Wilhelm and Manjunath, 2015]. Variances and scaling factors can be kept positive by using the log-exp transform. Unfortunately, the computation of the likelihood and its gradient can only be done for small models in practice, because the evaluation of the multivariate Gaussian CDF is time consuming, necessitating the use of sampling-based approximations. Our experiments refer to this as full MLE.

## 5.2 THE BLICA METHOD

However, we can circumvent the computational burden of the high-dimensional Gaussian cumulative distribution function. Due to the theory in Section 4, the correlations of $\mathbf{q}^u$ convey the essential information between the binary data and the continuous mixing model. Since the marginalization properties of our model are inherited from the multivariate Gaussian, such correlations can be estimated from *pairwise* marginal distributions of elements of $\mathbf{x}^u$; in 2D the Gaussian cumulative distribution function is still quite quick to compute. Thus, we combine maximum likelihood estimation with what could be called a "moment-matching" approach as follows. We first recover the pairwise correlations of the continuous-valued $\mathbf{q}^u$ from the observed binary data on $\mathbf{x}^u$ (this is possible by Corollary 1) via MLE in 2D. Then we fit those correlations to the correlations implied by the latent linear mixing model using a more scalable MLE in the continuous-valued latent space. The resulting algorithm is summarized as Algorithm 1 and explained in detail below.

**Correlation estimation.** On line 4, we estimate each correlations between elements in $\mathbf{q}^u$ separately, by directly fitting the likelihood in Equation 7 in two dimensions, thus estimating $\boldsymbol{\mu}_{\mathbf{q}}^u$ and $\mathbf{\Sigma}_{\mathbf{q}}^u$. To calculate the multivariate Gaussian CDF, we use the R package mvtnorm [Genz and Bretz, 2009]. We employ the GenzBretz method, which is particularly suitable for the fast evaluation needed here [Genz, 1993]. Furthermore, the estimation can be simplified [Lee and Sompolinsky, 1999]. Due to Equation 11 the diagonal of $\mathbf{\Sigma}_{\mathbf{q}}^u$ can be set to 1s in this step. Furthermore, since the marginal of

**Algorithm 1** The BLICA algorithm for Binary ICA.

---
1: Input data recorded at $n_u$ different segments.
2: **for** segment $u \in \{1, \ldots, n_u\}$ **do**
3:     **for** each observed variable pair $\{x_i^u, x_j^u\}$ **do**
4:         Estimate the correlation between $q_i^u$ and $q_j^u$ by maximizing the marginal pairwise likelihood of $x_i^u$ and $x_j^u$ (in segment $u$).
5:     Form and regularize the correlation matrix $\mathbf{C}_\mathbf{q}^u$ obtained from the pairwise correlations.
6: Optimize scaled Gaussian likelihood with L-BFGS over sufficient statistics $\mathbf{C}_\mathbf{q}^u$ from all segments $u$.
7: Return the estimated mixing matrix $\mathbf{A}$ and source variances $\mathbf{\Sigma}_\mathbf{z}^u$ for all segments $u$.

---

$x_i^u$ is

$$P(x_i^u = 1) = \Phi(-\boldsymbol{\mu}_\mathbf{q}^u[i]/\sqrt{\mathbf{\Sigma}_\mathbf{q}^u[i,i]}|0,1), \quad (12)$$

both means in $\boldsymbol{\mu}_\mathbf{q}^u$ can be computed from the respective marginals using the 1D inverse Gaussian CDF separately [Genz and Bretz, 2009]. The univariate optimization problem for the remaining parameter in the interval $[-1, 1]$ can then be solved efficiently using a line search method [Brent, 2013]. The scalability of Algorithm 1 depends crucially on this step, as $n_u \cdot (n^2 - n)/2$ correlations need to be estimated. The separately estimated correlations are collected to $n_u$ segmentwise $n$-by-$n$ correlation matrices denoted by $\mathbf{C}_\mathbf{q}^u$.

**Regularization.** When estimating the correlations of $\mathbf{q}^u$ from sample data, it can happen that a correlation matrix $\mathbf{C}_\mathbf{q}^u$ is close to singular or not positive definite. We use the following regularization on line 5, based on the parameter $r$ [Warton, 2008], which marks the approximate condition number targeted. The regularized correlation matrix is then

$$\frac{1}{1+\delta}(\mathbf{C}_\mathbf{q}^u + \delta\mathbf{I}), \text{ where } \delta = \max\left(0, \frac{\lambda_1 - r \cdot \lambda_n}{r - 1}\right),$$

where $\lambda_1$ is the largest and $\lambda_n$ the smallest eigenvalue of $\mathbf{C}_\mathbf{q}^u$. This regularization keeps the unit diagonal.

**Moment Matching.** Finally, on line 6, we fit the model parameters (including stationary $\mathbf{A}$) to the estimated correlations $\mathbf{C}_\mathbf{q}^u$ using a Gaussian likelihood model over the different segments $u$ (Section 2). But in contrast to the usual case where we have the covariance matrices, here we need to account for the "binarization indeterminacy", resulting in additional nuisance scaling parameters, as pointed out above. We use the term scaled Gaussian likelihood to refer to the ordinary multivariate Gaussian likelihood where we include additional parameters $\mathbf{Q}^u$ as the scaling factors. The fitting is thus done by the following scaled Gaussian likelihood based on the sufficient statistics $\mathbf{C}_\mathbf{q}^u$:

$$l = \sum_{u=1}^{n_u} \frac{N}{2} \left[ -\log(\det(\mathbf{\Sigma}_\mathbf{q}^u)) - \text{Tr}(\mathbf{C}_\mathbf{q}^u(\mathbf{\Sigma}_\mathbf{q}^u)^{-1}) \right]$$

where recall that $\mathbf{\Sigma}_\mathbf{q}^u = \mathbf{Q}^u(\mathbf{I} + \mathbf{A}\mathbf{\Sigma}_\mathbf{z}^u\mathbf{A}^T)\mathbf{Q}^u$ by Equation 11 is a function of the mixing matrix $\mathbf{A}$, source variances $\{\mathbf{\Sigma}_\mathbf{z}^u\}_u$ (diagonal, positive elements) and scaling factors $\{\mathbf{Q}^u\}_u$ (diagonal, positive elements). Variances and scaling factors can be kept positive by using the log-exp transform. Note that without the scaling factors $\{\mathbf{Q}^u\}_u$, the mixing matrix $\mathbf{A}$ could be found via joint diagonalization [Miettinen et al., 2017]. Note also that due to Theorem 3, the source means do not need to be estimated. Here, instead, we perform the fitting by maximizing this likelihood using L-BFGS [Liu and Nocedal, 1989] with respect to the aforementioned parameters.

### 5.3 BINARY ICA THROUGH LINEAR IVAE

Khemakhem et al. [2020] presented the identifiable Variational Autoencoder (iVAE), an approach for nonlinear ICA employing variational autoencoders [Kingma and Welling, 2014, Rezende et al., 2014] that assumes access to an additionally observed variable such that the sources are independent given the auxiliary variable; further, each source follows an exponential family distribution given the auxiliary variable. Here, we apply the iVAE approach to estimate the binary ICA model from Section 2 [Barin Pacela, 2021]. As proposed by Kingma and Welling [2014] and Khemakhem et al. [2020], we use the factorized Bernoulli observational model and apply a sigmoid function element-wise to the output of the decoder to obtain the binary probabilities. Due to the linearity of our mixing model and the segment-wise structure, we can simplify the encoder (posterior approximation) of the VAE, and make all the transformations in the iVAE affine or linear, thus greatly simplifying the system. The linear iVAE is presented in more detail in Appendix E.

### 5.4 ESTIMATION OF THE SOURCES

After estimating the mixing matrix $\mathbf{A}$, it may be desired to estimate the sources $\mathbf{z}^u$ as well. In the case of binary data, the individual source values cannot be accurately estimated (even up to scale and order indeterminacies) due to the inherent noise introduced by the binarization procedure. Presumably, though, if the number of observed variables is large and the number of sources is small, the estimation may be reasonable. In any case, the posterior $P(\mathbf{z}^u|\mathbf{x}^u)$ can be easily calculated after estimating the mixing matrix.

## 6 EXPERIMENTS

We implemented our proposed methods and baselines using R (BLICA, full MLE) and python (linear iVAE). Here we investigate the identifiability of the model, as well as the finite-sample estimation performance and the scalability of our methods, also comparing to previous approaches.

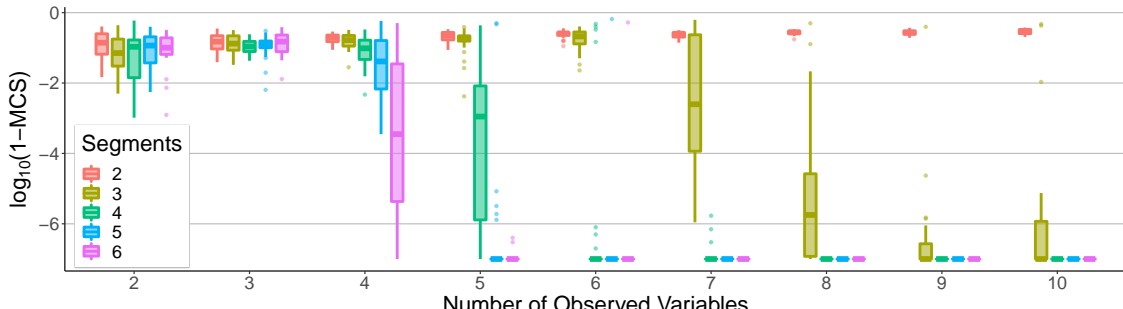

Figure 3: Identifiability with equal number of observed variables and sources. The BLICA method used true (pairwise) probability distributions (i.e. infinite sample limit data). Each box is based on 30 models. A lower value on the y-axis (log-error) implies better performance. Runs with values less than $-7$ (e.g. those in which the model was identified up to machine precision) are marked with $-7$. Compare to Table 1.

**Data.** The data was generated from the Binary ICA model (Section 2) in the following way. Means were drawn from $\text{unif}(-0.5, 0.5)$, standard deviations from $\text{unif}(0.5, 3)$. Mixing matrix elements were drawn from $\text{unif}(-3, 3)$ while ensuring invertibility by resampling until the condition number ($\kappa$) was below 20 for $n < 20$, or for $n \leq 20$ below the 75th quantile of 1000 sampled similar dimensional mixing matrices. For practical estimations from finite sample data we use 40 segments, varying the sample size per segment.

**Evaluation.** ICA methods are often compared in terms of the mean correlation coefficient of the estimated sources. Here, however, binarization induces heavy noise and individual samples of the estimated sources cannot be accurately estimated. We therefore focus our evaluation on the mixing model, and measure the mean cosine similarity (MCS) of the mixing matrix columns (taking the inherent order and scale indeterminacy of the sources into account, see Appendix D).

## 6.1 IDENTIFIABILITY

**Results.** Recall from Sections 4 and 5 that the correlations of $\mathbf{q}^u$ convey the information between the binary data and the mixing model, and each of these correlations can be determined from the marginal distributions over the corresponding pair of binary observed variables in $\mathbf{x}^u$ (in a segment $u$). Thus, by using the exact pairwise binary distributions of elements of $\mathbf{x}^u$ from Equation 6 as input for BLICA, we are here able to investigate identifiability empirically without any finite sample effects. Figure 3 shows results on which models can be identified when the number of sources and observed variables are equal ($n = n_z$). In many cases, the method found the mixing matrix essentially up to machine precision, which can be seen as indication of identifiability. Each box includes 30 different data generating models, and for each we ran BLICA 3 times; the MCS of the run with highest scaled Gaussian likelihood is plotted. With only 2 segments, or only 2 observed variables (also in

| Number of | Number of Observed Variables ($n$) | | | | | | | | |
|---|---|---|---|---|---|---|---|---|---|
| Segments ($n_u$) | 2 | 3 | 4 | 5 | 6 | 7 | 8 | 9 | 10 |
| 2 | -6 | -9 | -12 | -15 | -18 | -21 | -24 | -27 | -30 |
| 3 | -7 | -9 | -10 | -10 | -9 | -7 | -4 | **0** | **5** |
| 4 | -8 | -9 | -8 | -5 | **0** | **7** | **16** | 27 | 40 |
| 5 | -9 | -9 | -6 | **0** | 9 | 21 | 36 | 54 | 75 |
| 6 | -10 | -9 | -4 | **5** | 18 | 35 | 56 | 81 | 110 |

Table 1: Heuristic identifiability analysis. Each entry states the number of statistics (equations) minus the number of unknowns. The minimal cases with a non-negative number, suggesting identifiability, are bolded in red.

Theorem 3), the model is not identifiable in any case. The minimal cases deemed identifiable (up to source scale and order) are $(n=5, n_u=5)$, $(n=6, n_u=4)$, $(n=7, n_u=4)$, $(n=8, n_u=4)$, $(n=9, n_u=3)$, and $(n=10, n_u=3)$. Thus generally, the more observed variables ($n$) we have, the less segments ($n_u$) are needed.

**Heuristic Identifiability Analysis.** We contrast the results to the well-known heuristic approach to identifiability used in factor analysis. It is based on counting the number of statistics we can calculate (or equations we can form), and the number of unknowns (parameters) we need to solve. If the former is at least as large as the latter, there is hope that the model is identifiable. The calculations in Table 1 are based on Equations 10 and 11 when the number of sources equals the number of observations ($n = n_z$). The statistics correspond to $n_u(n^2 - n)/2$ covariances, $n_u \cdot n$ variances and $n_u \cdot n$ means (for $\mathbf{q}^u$). Unknowns include $n \cdot n$ mixing matrix coefficients, $n_u \cdot n$ (segment-wise) source variances, $n_u \cdot n$ source means, as well as $n_u \cdot n$ scaling terms (diagonal elements of $\mathbf{Q}^u$). In line with the classical literature in factor analysis, we ignore the source order indeterminacy. Figure 3 and Table 1 show a remarkably similar dependence between identifiability and the numbers of the segments and the observed variables: in particular, they agree on the mini-

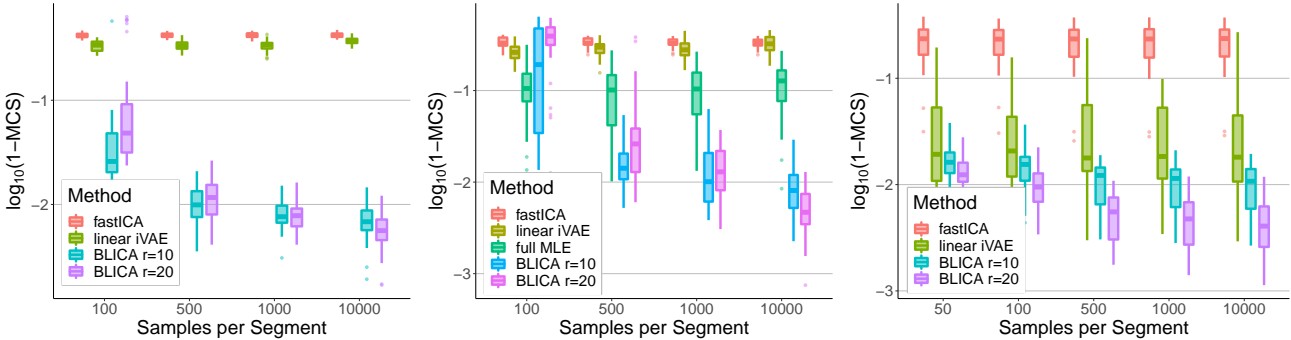

Figure 4: Finite sample performance. Left: 10 observed variables and 10 sources. Center: 6 observed variables and 6 sources. Right: 6 observed variables and 2 sources. Each box is based on thirty 40-segment datasets.

mal cases identifiable. Interestingly, cases with 2 observed variables as well as the cases with only 2 segments are never identifiable. Note that these computational results together with Section 4 provide a bound for any future analytical results on identifiability. If identifiability turns out to be possible in further cases, e.g., with a different mixing model or linking function, the results will need to depend on the particular parametric forms, thus limiting applicability.

## 6.2 FINITE SAMPLE ESTIMATION

**Methods.** Next we turn our attention to estimation performance from finite sample data. We compare our new BLICA (with different regularization parameter value $r$) method to its main competitors, fastICA [Himberg and Hyvärinen, 2001, Hyvärinen, 1999] and the baseline implementations of linear iVAE and full MLE. Note that the model of fastICA is somewhat different, but it still employs a linear mixing of the sources and has the same sources scale and order indeterminacies; thus, MCS comparison is sensible. fastICA does not use the segment index, but pools all data from different segments. Recall from Section 5.3 and Appendix E that the linear iVAE uses essentially the same model, but instead of employing the likelihood, it optimizes the ELBO objective through L-BFGS. For runs with $n < 20$ observed variables, a time budget of 2h was used, and the results that were obtained within the time limit are reported. For larger simulations, we allowed for 12h per run. To avoid local minima due to the difficult optimization landscape, we ran the linear iVAE, full MLE and BLICA with 3 different learning seeds and selected the best run according to the objective function (e.g. likelihood).

**Results.** Figure 4 (left) shows the result for 10 observed variables and 10 sources. BLICA clearly outperforms others consistently improving with increasing sample size. With smaller dimensions, 6 observed variables and 6 sources in Figure 4 (center), BLICA needs more samples to achive similar MCS. However, with fewer sources fewer samples are needed: Figure 4 (right) shows that for 6 observed variables

and 2 sources, high MCS can be obtained with only 50 samples per segments. Interestingly, linear iVAE performs well only with fewer sources than observations, while fastICA is not able to reliably estimate the mixing matrix from binary data. Unfortunately, full MLE cannot perform sufficiently many optimization steps within the time limit of 2h even with 6 observed variables in Figure 4 (center).

**Scalability.** Figure 5 assesses the performance in higher dimensions over data sets with 40 1000-sample segments, thirty for each $n$. Only BLICA can estimate the mixing matrix with equal number of observed variables equals and sources in Figure 5 (left). When the number of sources is fixed to 10 in Figure 5 (center), also linear iVAE shows improving performance with increasing number of observed variables. Finally, Figure 5 (right) shows the running time performance of BLICA (Algorithm 1) on the previous runs. The estimation of the quadratic number of correlations starts taking considerable time with 100 observed variables. L-BFGS is relatively quick in solving the optimization problem to a solution close to the final result (i.e. 1% lower MCS), then still gradually improving.

## 7 RELATED WORK

Our research connects particularly to the following earlier and more recent literature. Himberg and Hyvärinen [2001] consider binary observed vectors $\mathbf{x}$ and binary sources $\mathbf{z}$, so that the ICA mixing model is given by the Boolean expression $x_i = \bigvee_{j=1}^{n_z} a_{ij} \wedge z_j$. They show that this Boolean OR mixing can be approximated by a linear mixing model followed by a unit step function. Thus, they propose to estimate the model by ordinary ICA, and obtain reasonable results when the data is very sparse. Similarly, Nguyen and Zheng [2011] studied binary ICA with OR mixtures by defining a disjunctive generative model. They prove identifiability and propose an algorithm without continuous-valued approximations.

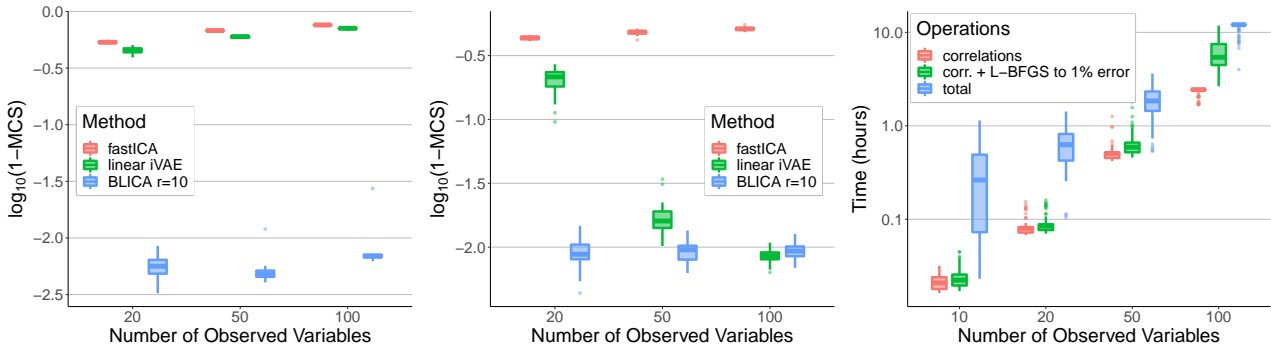

Figure 5: Scalability. Left: equal number of sources and observed variables. Center: 10 sources. Each box is based on thirty 40-segment datasets with 1000 samples per segment. Right: Running times of the steps of BLICA (Algorithm 1).

Kabán and Bingham [2006] proposed a model where continuous sources follow a Beta distribution, followed by a binary observation model. While their approach is related to ours, their latent variables are restricted to a finite interval, and they estimate the model using variational approximation which is unlikely to yield consistent estimators. Discrete ICA has further been approached by extensions of LDA where the topic intensities are mutually independent [Podosinnikova et al., 2015, Buntine and Jakulin, 2005, Canny, 2004]. Although their identifiability guarantees are limited [Podosinnikova et al., 2016], their method has the advantage of allowing for discrete data. Lee and Sompolinsky [1999] consider PCA employing a binarized Gaussian model.

Finally, we note that the very idea of estimating latent variable models by non-stationarity, originating in [Matsuoka et al., 1995, Pham and Cardoso, 2001], has been recently increasingly used in estimating generative models [Hyvärinen and Morioka, 2016, Khemakhem et al., 2020] as well as for causal discovery [Zhang et al., 2017, Monti et al., 2019], even in deep learning. Automatically estimating the segment index by a HMM has been further proposed by Hälvä and Hyvärinen [2020]. Instead of the wide-spread idea of joint diagonalization of covariance matrices [Belouchrani et al., 1997, Tsatsanis and Kweon, 1998], we used correlation matrices without explicit diagonalization criteria; related work on diagonalizing correlation matrices can be found in [Joho and Rahbar, 2002].

## 8 CONCLUSION

We presented a model for ICA of binary data which is based on a linear latent mixing model and non-stationarity of the sources. We investigated the identifiability, showing some surprising indeterminacies not present in ordinary ICA, including the fact that in the two-variable case the model cannot be identified. We believe that our identifiability results, theoretical and empirical, will be useful in future research on binary ICA. Based on our approach using a Gaussian link function, the likelihood can be obtained in closed form

although the Gaussian cumulative distribution function is still computationally heavy. These advances allowed for a practical method BLICA that combines maximum likelihood estimation and moment-matching; it was shown to be applicable in higher dimensions while still empirically showing consistent behaviour. As future work, we aim to generalize from binary to discrete variables, consider parallelized approaches for scaling up full MLE estimation, and investigate the potential of the new learning algorithm in applications.

## Acknowledgements

The first author was supported by the Academy of Finland under grant 315771. The second author acknowledges funding from Samsung Electronics Co., Ltd. (at Mila). The third author acknowledges funding from the Academy of Finland and a CIFAR Fellowship.

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
