# OpenReview forum: "Binary Independent Component Analysis: A Non-stationarity-based Approach"
_auai.org/UAI/2022/Conference — UAI 2022 Poster_

### Official Review · Reviewer_8b6x · 2022-04-12

**Q2(1) Originality/Novelty:** 2
**Q2(2) Significance/Impact:** 2
**Q2(3) Correctness/Technical Quality:** 3
**Q2(6) Clarity Of Writing:** 3
**Q6 Overall Score:** 5
**Q8 Confidence In Your Score:** 3

**Q1 Summary And Contributions:**

The authors propose a new model for independent component analysis of binary signals, by imposing a multivariate Gaussian distribution on the latent source signal. A thorough theoretical analysis of model's identifiability is conducted. A closed form for the model's estimation is available and can be used for MLE estimation. An alternative, computationally efficient, approximate estimation procedure, called BLICA, is presented. The BLICA performance is compared through a simulation study.

**Q2 Assessment Of The Paper:**

More detailed information regarding each of these aspects is given below:

**Q2(4) Quality Of Experiments (Optional):**

4: Excellent: The experimental evaluation is comprehensive and the results are compelling.

**Q2(5) Reproducibility:**

4: Excellent: Key resources (e.g., proofs, code, data) are available and key details (e.g., proof sketches, experimental setup) are comprehensively described for competent researchers to confidently and easily reproduce the main results.

**Q3 Main Strengths:**

The paper is well written with clear motivation and goals. The presented model is simple but provides a sound answer to a practical problem. The simulation study is convincing.


**Q4 Main Weakness:**

Some part of the methodology would benefit from from further explanation, especially how the non-stationary part is handled. Also, the "regularization" step of the BLICA algorithm would benefit from further justification. More generally, it is better from a scientific perspective to discuss "related work" at the beginning of a contribution and not at the end.


**Q5 Detailed Comments To The Authors:**

As explained in the previous section, it would be beneficial to give more details about the "u" component. It might be clear to a reader familiar with the literature in the field, but not to the general audience such as the UAI community. For instance, two major points could be clarified: in practice, are the segments pre-defined or do you need to estimate them from the data before applying ICA? How does the non-stationnarity increase the model identifiability?

On a related topic, Figure 3 is difficult to understand: it would be nice to have a short sentence recalling that lower value of $log_10(1-MCS)$ implies better performance and that a model is considered as identifiable for values below, say $-3$.

Some details about BLICA could be better explained / justified:
- the full MLE approach does not seem intractable for the dimensionalities considered in the paper. It might be necessary to use parallelization for reasonable computing time. This might be impractical but would be very interesting to compare the performance loss by BLICA.
- also for the full MLE, how would you parametrize the correlation matrix to ensures that its estimates satisfies the necessary properties?
- for BLICA, the so called "regularization" step seems a bit awkward. First, why is it called "regularization"?  In general in ML, this term refers to penalty terms penalizing for the model complexity while in this context, it seems to be closer to an attempt to project on the closest positive definite matrix. Second, the pairwise estimates do not ensure at all that the matrix is well defined, and that is the reason why such "regularization" is needed. Could the authors elaborate on why, beyond empirical evidence, this estimate is consistent?


Minor comments:
- p 2 c.1: use $\left( \text{ and } \right)$ in equations.
- p.5 c.1: "then we fit those correlation": what do you mean?
- p.6 c.1: maybe add a sentence to say that iVAE is introduced to be used as benchmark reference.

**Q7 Justification For Your Score:**

The paper is well written with compelling ideas and solution. Some parts would however benefit from further clarifications.


**Q9 Complying With Reviewing Instructions:**

1: Yes.

---

### Official Review · Reviewer_zKJ9 · 2022-04-12

**Q2(1) Originality/Novelty:** 3
**Q2(2) Significance/Impact:** 3
**Q2(3) Correctness/Technical Quality:** 3
**Q2(6) Clarity Of Writing:** 3
**Q6 Overall Score:** 6
**Q8 Confidence In Your Score:** 2

**Q1 Summary And Contributions:**


The paper tries to estimate parameters in a  linear mixing model in a continuous-valued latent space based on the binary observations.  The authors discuss the identifiability  of the proposed model and introduce methods to estimate the mixing matrix A and covariance matrix of  Independent Components.

**Q2 Assessment Of The Paper:**

More detailed information regarding each of these aspects is given below:

**Q2(4) Quality Of Experiments (Optional):**

3: Good: The experimental evaluation is adequate, and the results convincingly support the main claims.

**Q2(5) Reproducibility:**

3: Good: Key resources (e.g., proofs, code, data) are available and key details (e.g., proofs, experimental setup) are sufficiently well-described for competent researchers to confidently reproduce the main results.

**Q3 Main Strengths:**

Independent Component Analysis via Non-stationarity is an important issue.  The   identifiability  of the proposed model are discussed in detail and the proposed MLE is efficient.

**Q4 Main Weakness:**

Why use a specific link function  like, \Phi(\sqrt{\pi\over 8} y|0,1)?    It seems that this setting  is critical to the identifiability of the proposed model.  More motivation should be given about this setting.


**Q5 Detailed Comments To The Authors:**

~

**Q7 Justification For Your Score:**

Equal weighs

**Q9 Complying With Reviewing Instructions:**

1: Yes.

---

### Official Review · Reviewer_7yZz · 2022-04-12

**Q2(1) Originality/Novelty:** 3
**Q2(2) Significance/Impact:** 3
**Q2(3) Correctness/Technical Quality:** 2
**Q2(6) Clarity Of Writing:** 3
**Q6 Overall Score:** 6
**Q8 Confidence In Your Score:** 3

**Q1 Summary And Contributions:**

The authors proposed a binary ICA model which is based on a linear latent mixing model in a continuous space and non-stationarity of the sources. They showed although such a model can not be identified in the two-variable case, it can theoretically identify the correlation matrices for a defined random vector $\mathrm{q}^u$. They presented three estimation methods for binary ICA. Experiments showed their method’s empirical identifiable results when the number of observed variables is higher.

**Q2 Assessment Of The Paper:**

More detailed information regarding each of these aspects is given below:

**Q2(4) Quality Of Experiments (Optional):**

3: Good: The experimental evaluation is adequate, and the results convincingly support the main claims.

**Q2(5) Reproducibility:**

2: Fair: Key resources (e.g., proofs, code, data) are unavailable but key details (e.g., proof sketches, experimental setup) are sufficiently well-described for an expert to confidently reproduce the main results.

**Q3 Main Strengths:**

1. This paper is well written and the authors are good at providing intuitive examples for further explanation. And the binary ICA problems they focused on, including the identifiability and estimation methods, are important and potentially useful.

2. The authors found the non-identifiability for the binary ICA model in the two-variable case, which is somewhat surprising, but they showed empirically that the model is identifiable when the dimensionality is higher. Further, they employed correlation identifiability to derive a practical algorithm for the estimation.

**Q4 Main Weakness:**

I think overall the authors did interesting research, but I have some concerns listed below.

Since the title of this paper is “Binary Independent Component Analysis via Non-stationarity”, I think that the authors would have used some information about non-stationarity (e.g., the invariance) to help estimate such a non-stationary model, but the authors do not follow this direction. It seems to me that the segment variable $u$ (or the number of segments $n_u$) is given, as shown especially in Algorithm 1. And the authors only estimate the binary ICA model in one segment by another one. Thus it would confuse me whether they focus on handling the non-stationarity problem or not.



**Q5 Detailed Comments To The Authors:**

I have some concerns listed below,
- What are the relationships between the identifiability for the binary causal discovery model and the identifiability for the binary ICA model? It might be helpful to discuss more the applications of the binary ICA model in the paper.

- In Page 2, “add independent noise $\epsilon $ form $\mathcal{N}$” -> “add independent noise from $\mathcal{N}$”.

**Q7 Justification For Your Score:**

The binary ICA problems, including the identifiability and estimation methods, are important and potentially useful. And the authors did find some surprising and interesting results. I would increase my score if the main concerns are addressed.

**Q9 Complying With Reviewing Instructions:**

1: Yes.

---

### Official Review · Reviewer_MVkk · 2022-04-13

**Q2(1) Originality/Novelty:** 3
**Q2(2) Significance/Impact:** 3
**Q2(3) Correctness/Technical Quality:** 3
**Q2(6) Clarity Of Writing:** 3
**Q6 Overall Score:** 7
**Q8 Confidence In Your Score:** 4

**Q1 Summary And Contributions:**

In this paper, the authors consider the independent component analysis of binary data setting, in which the mixing model is linear. The authors provide identifiability and propose a new method for their model. The experimental results demonstrate the effectiveness of their approach.

**Q2 Assessment Of The Paper:**

More detailed information regarding each of these aspects is given below:

**Q2(4) Quality Of Experiments (Optional):**

3: Good: The experimental evaluation is adequate, and the results convincingly support the main claims.

**Q2(5) Reproducibility:**

3: Good: Key resources (e.g., proofs, code, data) are available and key details (e.g., proofs, experimental setup) are sufficiently well-described for competent researchers to confidently reproduce the main results.

**Q3 Main Strengths:**

The paper tackles a very challenging problem and provides a novel approach.

The authors have an in-depth understanding of the related works and provide a detailed review.

The theoretical contributions of this paper are solid, and the experiments are quite thorough.


**Q4 Main Weakness:**

The assumption of binary data seems a bit strict.

**Q5 Detailed Comments To The Authors:**

Non-Stationarity seems to be the most critical foundation of this paper and is worth more explanation and intuition like "what is the connection between the number of segments and the number of observed/latent variables for model identification".


In the experiments, what is the reason that you only run 3 times for each case? I think one may need to provide the computational complexity of the proposed algorithm.

Could you give more discussion about how to extend your method to address the discrete setting?


**Q7 Justification For Your Score:**

The problem of this paper is challenging and non-trivial. The proposed method is novel, and the theoretical guarantee for their method is solid.

**Q9 Complying With Reviewing Instructions:**

1: Yes.

---

### Decision · Program_Chairs · 2022-05-15

**Decision:**

Accept (Poster)

**Comment:**

Meta Review: I had trouble with this paper and I have to say that I am more skeptical than the reviewers, who were generally positive. Some of the concerns were raised by the reviewers and everybody seemed happy after the rebuttal so I will not push this any further, although, I expect that the authors can clarify considerable, e.g. on using segments in their set-up.

My main concern is that the title is misleading in many ways. First, it suggests that non-stationarity is handled in some special way in this paper but it is not. Second, such a general title "binary ICA" suggests that they came up with a canonical way of dealing with ICA for binary data. However, they approach is very special. The choice of this specific link function is not well motivated in the paper (it is in the replies). I would add to that that in the classical ICA the mixing matrix has a concrete physical interpretation but here such an interpretation is missing. This is of course an easily fixable concern and I hope the authors will adjust their title.

My other concern is about the identifiability results. I am not saying that the results are wrong but that the authors do not have a good understanding of the identifiability issue in this particular scenario and here the paper looks underdeveloped. But after addressing the comments of the reviewers and a title adjustment, I think this could be an interesting paper.